# Protocol for a Multi-Center Confirmatory Trial to Evaluate the Differential Diagnostic Performance of Contrast-Enhanced Ultrasonography Using Perflubutane in Patients with a Pancreatic Mass: A Multicenter Prospective Study

**DOI:** 10.3390/diagnostics14020130

**Published:** 2024-01-06

**Authors:** Yasunobu Yamashita, Toshio Shimokawa, Reiko Ashida, Yoshiki Hirooka, Takuji Iwashita, Hironari Kato, Toshifumi Kin, Atsushi Masamune, Haruo Miwa, Eizaburo Ohno, Hideyuki Shiomi, Atsushi Sofuni, Mamoru Takenaka, Masayuki Kitano

**Affiliations:** 1Second Department of Internal Medicine, Wakayama Medical University, 811-1 Kimiidera, Wakayama 641-0012, Japan; 2Clinical Study Support Center, Wakayama Medical University Hospital, Wakayama 641-0012, Japan; 3Department of Gastroenterology and Gastroenterological Oncology, Fujita Health University School of Medicine, Toyoake 470-1192, Japan; 4First Department of Internal Medicine, Gifu University Hospital, Gifu 501-1194, Japan; 5Department of Gastroenterology and Hepatology, Okayama University Graduate School of Medicine, Okayama 700-8558, Japan; 6Center for Gastroenterology, Teine Keijinkai Hospital, Sapporo 006-8555, Japan; 7Division of Gastroenterology, Tohoku University Graduate School of Medicine, Toyoake 980-8574, Japan; 8Gastroenterological Center, Yokohama City University Medical Center, Yokohama 232-0024, Japan; 9Department of Gastroenterology and Hepatology, University Graduate School of Medicine, Nagoya 466-8550, Japan; 10Division of Gastroenterology and Hepatobiliary and Pancreatic Diseases, Department of Internal Medicine, Hyogo Medical University, Nishinomiya 663-8501, Japan; 11Department of Gastroenterology and Hepatology, Tokyo Medical University, Tokyo 160-0023, Japan; 12Department of Gastroenterology and Hepatology, Kindai University Faculty of Medicine, Osaka 589-8511, Japan

**Keywords:** contrast-enhanced transabdominal ultrasonography, contrast-enhanced endoscopic ultrasonography, pancreatic cancer, perflubutane

## Abstract

For pancreatic masses, an evaluation of their vascularity using contrast-enhanced ultrasonography can help improve their characterization. This study was designed to evaluate the utility and safety of contrast-enhanced transabdominal ultrasonography (CE-TUS) and endoscopic ultrasonography (CE-EUS) in the diagnosis of pancreatic masses including solid or cystic masses. This multi-center comparative open-label superiority study is designed to compare Plain (P)-TUS/EUS alone with P-TUS/P-EUS plus CE-TUS/CE-EUS. Three hundred and one patients with a total of 232 solid pancreatic masses and 69 cystic masses were prospectively enrolled. The primary endpoints are to compare the diagnostic accuracy between P-TUS/P-EUS alone and P-TUS/P-EUS plus CE-TUS/CE-EUS for both the TUS and EUS of solid pancreatic masses, and to compare the diagnostic accuracy between P-EUS alone and P-EUS plus CE-EUS in cystic pancreatic masses. The secondary endpoints are to compare the diagnostic sensitivity and specificity of P-TUS/P-EUS alone and P-TUS/P-EUS plus CE-TUS/CE-EUS for pancreatic solid/cystic masses, and the accuracy of P-TUS alone and P-TUS plus CE-TUS for pancreatic cystic masses. Other secondary endpoints included comparing the diagnostic sensitivity, specificity, and accuracy of CE-TUS, CE-EUS and CE-computed tomography (CT) for solid/cystic pancreatic masses. The safety, degree of effective enhancement, and diagnostic confidence obtained with CE-TUS/CE-EUS will also be assessed.

## 1. Introduction

Pancreatic cancer is the fourth leading cause of cancer-related death in the USA and Japan, and the incidences of pancreatic cancer and related mortality have recently increased [1,2]. The National Cancer Center Japan reported that 37,677 people died of pancreatic cancer in Japan in 2020 [2]. Despite recent advances in diagnostic imaging modalities, most cases of pancreatic cancer are discovered at an unresectable stage for which the prognosis is poor, with a 5-year survival rate of approximately 11% [1]. This poor prognosis is mainly related to the difficulties associated with the diagnosis of pancreatic cancers at an early stage; early diagnosis is essential to ensure curative treatment and improve the prognosis of patients with pancreatic cancer [3].

For early stage I pancreatic cancer, Kanno et al. reported tumor detection rates of 67.3% for transabdominal ultrasonography (TUS), 65.8% for computed tomography (CT), 57.5% for magnetic resonance imaging (MRI), and 92.4% for endoscopic ultrasonography (EUS) [3]. Although TUS showed a similar diagnostic ability to CT and MRI for early pancreatic cancer, TUS is the simplest, most affordable, and most widely used modality, and can therefore be considered the most suitable modality for screening for pancreatic cancer. However, compared with other imaging modalities, including TUS, EUS provides superior spatial resolution for the early diagnosis of pancreatic cancer [4]. Despite this, plain TUS/EUS alone is limited in its ability to distinguish pancreatic cancer from non-neoplastic pancreatic masses because most solid pancreatic masses are detected as hypoechoic masses.

Pancreatic cystic tumors are frequency detected by cross-sectional imaging. Pancreatic cystic tumors can range from malignant to benign. In cystic tumors, intraductal papillary mucinous neoplasms (IPMNs) and mucinous cystic neoplasms (MCNs) have malignant potential. IPMNs are pancreatic cystic tumors that display a dilatation of the excretory pancreatic ducts and mucous production by papillary proliferation of the ductal epithelium. They commonly occur in the elderly. IPMNs are divided into the branch duct type, main pancreatic duct type, and mixed type. According to previous reports, the malignancy for main duct-type tumors ranges from 57% to 92% [5,6,7,8,9]. On the other hand, the malignancy rate for branch duct-type tumors is relatively low (6–46%) [10,11,12,13]. Therefore, in branch duct-type IPMNs, an assessment of the mural nodules is important for malignant transformation. Therefore, the presence of mural nodules ≥5 mm in size is identified as an important factor for making decisions on surgical intervention in the 2017 guidelines for IPMN [14]. MCNs are mucus-producing cystic tumors that are common in middle-aged women. All MCNs should be resected to prevent malignant changes. According to a previous report, the malignancy is 17.3% in resected MCNs [15]. On the other hand, serous cyst neoplasms (SCNs) are multilocular cystic tumors with serous content that consist of glycogen-rich clear cells. SCNs are generally considered benign. Therefore, differential diagnosis is important. On the other hand, EUS is one of the most reliable and efficient diagnostic modalities for pancreatic cystic tumors due to its superior spatial resolution compared with any other modality. However, it is sometimes difficult to diagnose markers of malignant transformation such as the presence of a mural nodule.

In this context, an assessment of the vascularity of masses is useful for the differential diagnosis of pancreatic masses detected on P-TUS/P-EUS. In clinical practice, contrast-enhanced TUS/EUS (CE-TUS/CE-EUS) has become increasingly used as an adjunctive method for the characterization of pancreatic tumors. Moreover, there are three advantages of CE-EUS over the other imaging methods. First, CE-TUS/CE-EUS can detect signals from microbubbles in vessels with a very slow flow and without Doppler-related artifacts compared to contrast-enhanced color and power Doppler TUS/EUS. Second, CE-TUS/CE-EUS has an advantage over contrast-enhanced MRI and CT in patients with contraindications in the use of contrast materials, such as in renal failure or contrast allergy, even though adverse reactions to contrast agents are rare in humans. Third, CE-TUS/CE-EUS allows real-time dynamic imaging and repeated examinations.

The underlying principle of contrast harmonic imaging is as follows: when microbubbles in the contrast agent are exposed to the ultrasound beams, they are disrupted or resonate, releasing many harmonic signals. When the tissue and microbubbles receive transmitted ultrasound waves, both produce harmonic components that are integer multiples of the fundamental frequency. A selective depiction of the second harmonic component visualizes signals from microbubbles, which are stronger than those from tissue because the harmonic content from microbubbles is higher than that from tissue [16]. Therefore, this technology can detect signals from microbubbles in vessels with a very slow flow without Doppler-related artifacts and is used to characterize vascularity [17].

There are many reports on the differential diagnosis of pancreatic masses using CE-TUS/CE-EUS, including solid and cystic masses [18,19,20,21,22,23,24,25,26,27,28,29,30,31,32,33,34,35,36,37,38,39,40,41,42,43,44,45,46,47,48,49,50,51,52,53,54,55,56,57,58,59,60,61]. However, the criteria for diagnosing malignancy in solid masses differ considerably from those for cystic masses, and it is therefore necessary to evaluate solid and cystic masses separately. For solid masses, it was reported that CE-TUS/CE-EUS is useful for the differential diagnosis of pancreatic cancer when a hypoenhancement pattern on CE-TUS/CE-EUS is defined as malignancy [20,21,22,23,24,26,27,28,29,30]. When a hypoenhancement tumor on CE-EUS and irregular contour on P-EUS was defined as malignancy, CE-EUS was significantly more accurate than P-EUS (respectively, sensitivity 90.8% vs. 95%; specificity; 74.6% vs. 42.9%; accuracy 85.8% vs. 78.9%, *p* < 0.001) [21]. Although there are several criteria for determining malignant cysts, including nodule morphology, vessel pattern in the nodule, and the presence of enhancement in a nodule [31,32,33,34,35], the presence of enhancement in a nodule may be preferred as the criterion because diagnostic criteria using the presence of enhancement in a nodule among these three kinds of criteria are simple and objective. It was reported that, in cystic masses, CE-EUS is useful for determining malignancy when a mass with an enhanced mural nodule on CE-EUS is defined as malignancy [31,34,35]. When the presence of mural nodule was defined as malignancy, CE-EUS was significantly more accurate than P-EUS (respectively, sensitivity 97% vs. 97 %; specificity; 75% vs. 40%; accuracy 84% vs. 64%, *p* = 0.0001) [31].

Although there are many retrospective studies on CE-TUS/CE-EUS, there are few prospective studies including solid/cystic masses. Therefore, we planned this prospective trial to confirm the advantages of P-TUS/P-EUS plus CE-TUS/CE-EUS over P-TUS/P-EUS alone in the diagnosis of pancreatic masses, including solid/cystic masses.

## 2. Methods and Design

### 2.1. Data Collection

Data are collected prospectively from all patients, and includes history, physical examination, laboratory data, pathological examination, clinical information, and adverse events. The study allocation, intervention, and assessment are adapted from standard protocol items (Figure 1).

### 2.2. Data Monitoring and Audit

The following aspects are monitored: data accumulation, patient eligibility, severe adverse events, protocol deviations, reasons for cessation or expiration of the protocol, background factors of the patients, and other problems concerning study progress and safety.

All study documentation and the source data/documents are accessible to auditors/inspectors, and questions are answered during inspections.

### 2.3. Images

To standardize the setting of processors and image interpretation, several discussions were held using movie clips of P-TUS/P-EUS and CE-TUS/CE-EUS before the initiation of the study. For P-TUS/P-EUS, tissue harmonic TUS/EUS is used for plain images. For CE-US/CE-EUS, the mechanical index is set at 0.18–0.2 and 0.2–0.4, respectively. After reconstitution with 2 mL of sterile water for injection, 0.015 mL/kg of the contrast agent is administered through a peripheral vein.

CE-CT is performed with the following settings: 64-row CT or more CT scanner, slice width ≤ 5 mm, and at least three phases (pancreatic parenchymal, portal, and delayed phases).

### 2.4. Final Diagnosis

The final diagnosis will be based on pathological findings obtained by EUS-FNA and/or surgery when the pathological diagnosis is malignant.

When pathological diagnosis is not malignant, the final diagnosis will be based on comprehensive diagnosis with imaging findings excluding CE-TUS/CE-EUS imaging, tumor markers, and pathological diagnosis.

### 2.5. Definitions

The diagnostic criteria were agreed following a discussion between 10 experts from 9 institutions before the clinical trial.

#### 2.5.1. Pancreatic Solid Mass

##### On P-TUS/P-EUS

A heterogeneous hypoechoic mass with an irregular contour on P-TUS/P-EUS is defined as malignancy.

##### On CE-TUS/CE-EUS

A hypoenhancement pattern on CE-US, in which the echo intensity of the mass is lower than that of the surrounding pancreatic tissue, is defined as malignancy.

##### On CE-CT

A hypovascular nodule on CE-CT is defined as malignancy.

#### 2.5.2. Cystic Mass

##### On P-US/P-EUS

A mass with a mural nodule on P-TUS/P-EUS is defined as malignancy.

##### On CE-TUS/CE-EUS

A mass with an enhanced mural nodule on CE-TUS/CE-EUS is defined as malignancy.

##### On CE-CT

A mass with an enhanced mural nodule on CE-CT is defined as malignancy.

### 2.6. Image Review

During blind reading, P-TUS/CE-TUS, P-EUS/CE-EUS, and CE-CT data are reviewed independently in a random order by 3 expert readers for each modality.

### 2.7. Primary Endpoint

The primary endpoints are to compare the diagnostic accuracies of P-TUS/P-EUS alone and P-TUS/P-EUS plus CE-TUS/CE-EUS for the diagnosis of pancreatic solid masses, and P-EUS/CE-EUS for cystic masses.

### 2.8. Secondary Endpoints

The secondary endpoints are comparisons of the diagnostic sensitivity and specificity of P-TUS/P-EUS alone and P-TUS/P-EUS plus CE-US/CE-EUS for pancreatic solid/cystic masses, and the accuracy of P-TUS alone and P-TUS plus CE-TUS for pancreatic cystic masses. Other secondary endpoints are comparisons of the diagnostic sensitivity, specificity, and accuracy of CE-TUS/CE-EUS and CE-CT for pancreatic solid/cystic masses. The safety, degree of effective enhancement, and diagnostic confidence obtained with the contrast agents will also be assessed.

### 2.9. Statistical Analysis

The assessments will be performed under fallback procedures. First, we will examine the superiority of the use of the contrast agent in EUS in the evaluation of the accuracy of the differential diagnosis of pancreatic cancer and non-pancreatic cancer, making comparisons with non-use of the contrast agent. This means that a one-sided alternative hypothesis H11: “the accuracy of CE-EUS for the differential diagnosis of pancreatic cancer and non-pancreatic cancer (accuracy rate) is higher than that of P-EUS” will be evaluated against the null hypothesis H01: “the accuracy of CE-EUS for differential diagnosis of pancreatic cancer and non-pancreatic cancer is equivalent to that of P-EUS.” This comparison will be made using McNemar’s test with a significance level of α1 = 0.0125. If the *p*-value is lower than the significance level α1, a significance level α2 for the assessment of the accuracy of CE-EUS in the differential diagnosis of pancreatic cancer and non-pancreatic cancer will be established as 0.025. By contrast, if the *p*-value is not less than 0.0125, a significance level α2 will be established as 0.0125. To determine the superiority of contrast agent use compared with non-use in the evaluation of the accuracy of CE-EUS for the differential diagnosis of pancreatic cancer and non-pancreatic cancer, a one-sided alternative hypothesis H12: “the accuracy of CE-TUS for the differential diagnosis of pancreatic cancer and non-pancreatic cancer is higher than that of P-TUS” will be evaluated against the null hypothesis H02: “the accuracy of CE-TUS for differential diagnosis of pancreatic carcinoma and non-pancreatic carcinoma is equivalent to that of P-TUS.” This evaluation will be made using McNemar’s test with a significance level of α2. If only the null hypothesis H01 is rejected in the above assessment, the results will be interpreted as meaning that the use of the contrast agent increases the accuracy of CE-EUS alone for the differential diagnosis. If only the null hypothesis H02 is rejected, it will be interpreted as meaning that the use of the contrast agent increases the accuracy of CE-TUS for the differential diagnosis. If both of the null hypotheses are rejected, it will be interpreted as meaning that the use of the contrast agent increases the accuracy of both CE-TUS and CE-EUS for the differential diagnosis. If any of the null hypotheses are not rejected, it will be interpreted as meaning that it cannot be said that the use of the contrast agent increases the accuracy of either CE-TUS or CE-EUS for the differential diagnosis (Figure 2).

### 2.10. Sample Size Calculation

#### 2.10.1. Pancreatic Solid Masses

The objective of this trial is to verify (1) that the accuracy of CE-EUS for the differential diagnosis of pancreatic cancer and non-pancreatic cancer is higher than that of P-EUS (superiority), and (2) that the accuracy of CE-TUS for the differential diagnosis of pancreatic cancer and non-pancreatic cancer is higher than that of non-contrast-enhanced P-TUS (superiority). To verify these co-primary endpoints, the fallback procedure in the gate-keeping strategy will be used.

Omoto et al. [21] reported diagnostic accuracy rates of 87.3% CE-EUS (with the use of the contrast agent plus non-use of the contrast agent; Combination) and P-EUS without the use of the contrast agent. According to the report, the accuracy rate was 87.3% for the former procedure and 78.9% for the latter procedure. Based on these results, a calculation was made, and the results showed that the proportion of a correct diagnosis by combination and an incorrect diagnosis by P-EUS was 10.3% (21/204 subjects) and the proportion of an incorrect diagnosis by combination and a correct diagnosis by P-EUS was 2.0% (4/204 subjects). It is assumed that similar results can be obtained in both TUS and EUS. To assure the familywise error rate (FWER) of 0.025 (one-sided alternative hypothesis), 0.025/2 = 0.0125 is assigned to Objective (1) and Objective (2).

Under the above setting, when the McNemar test is run on paired samples of the use and non-use of the contrast agent with a significance level of α = 0.0125 (one-sided) and 185 subjects as the minimum required number of subjects, the power 1-β is 0.840. For the fallback, if Objective (1) is significant, Objective (2) is assessed with a significance level of α = 0.025 (one-sided). In this case, the power 1-β is 0.902. In this study, the target sample size will be set at 232 subjects on the assumption that approximately 20% of subjects may be ineligible or not evaluable. The power 1-β under the target sample size of 232 subjects is 0.924 with a significance level of α = 0.0125 (one-sided), and 0.955 with a significance level of α = 0.0250 (one-sided).

#### 2.10.2. Cystic Masses

Kamata et al. [31] reported diagnostic results for pancreatic cystic masses on P-EUS and CE-EUS. According to their report, the accuracy rate was 64.2% for P-EUS and 84.3% for CE-EUS. A calculation was made on the basis of these results that showed that the proportion showing a correct diagnosis with use of the contrast agent and an incorrect diagnosis without use of the contrast agent was 21.4% (15 subjects/70 subjects), and that the proportion showing an incorrect diagnosis with use of the contrast agent and a correct diagnosis with non-use was 1.4% (1 subject/70 subjects). It is assumed that similar results will be obtained in this clinical trial. Under the above setting, in patients with a pancreatic cystic mass, when the McNemar’s test is run for the use and non-use of the contrast agent with a significance level of α = 0.0250 (one-sided) and 55 subjects as the minimum required number of subjects, the power 1-β will be 0.904. Assuming that approximately 20% of subjects may be ineligible, the target sample size will be 69 subjects. With a sample size of 69 subjects, the power 1-β is 0.965.

## 3. Discussion

This multi-center comparative open-label superiority study is designed to compare P-TUS/P-EUS alone and P-TUS/P-EUS plus CE-TUS/CE-EUS in the diagnosis of pancreatic masses including solid and cystic masses.

In a meta-analysis of CE-TUS for pancreatic solid masses, the pooled estimates of sensitivity, specificity, and area under the summary receiver operating characteristics curve (AUC) for the diagnosis of pancreatic cancer were 92% (95% confidence interval [CI].: 0.89–0.94), 76% (95% CI: 0.71–0.81), and 0.95, respectively [56]. In comparisons of CE-TUS and CE-CT, CE-TUS was found to be superior to CE-CT in three reports [57,58,59]. In comparisons of P-TUS and CE-TUS, CE-TUS was found to be superior to P-TUS for pancreatic cancer [60]. However, we are not aware of any reports comparing P-TUS with P-TUS plus CE-TUS.

In a meta-analysis of CE-EUS for pancreatic solid masses, the pooled estimates of sensitivity, specificity, and AUC for the diagnosis of pancreatic cancer were 93% (95% CI, 0.91–0.95), 80% (95% CI, 0.75–0.85), and 0.97, respectively [18]. In three reports comparing CE-EUS and CE-CT, CE-EUS was found to be superior to CE-CT [23,24,29]. In two of these three studies, CE-EUS was also found to be superior to CE-CT for diagnosing small pancreatic solid masses [23,24]. There is only one report comparing P-EUS and P-EUS plus CE-EUS for pancreatic cancer [21], and we therefore used the results of this report to estimate the sample size for this trial.

Although the utility of CE-TUS with the first-generation contrast agent galactose-palmitic acid (Bayer Yakuhin Ltd., Osaka, Japan) was reported, there are no reports on CE-TUS with perflubutane for the diagnosis of pancreatic cystic masses.

In a meta-analysis on CE-EUS, the pooled estimates of sensitivity, specificity, and diagnostic accuracy for malignant cysts (when malignancy was defined as the presence of hyperenhancement on CE-EUS) were 97.0% (95% CI, 92.5–99.2%), 90.4% (95% CI, 85.2–94.2%), and 95.6% (95% CI, 92.6–98.7%), respectively [61]. There is only one report comparing CE-EUS with CE-CT in the diagnosis of malignant cysts [32] and this report showed that CE-EUS was superior to CE-CT for the diagnosis of malignant cysts. Three other reports showed CE-EUS to be superior to P-EUS for malignant cysts [31,34,35].

If this trial shows P-TUS/P-EUS plus CE-TUS/CE-EUS to be superior to P-TUS/P-EUS alone, CE-TUS/CE-EUS may bring about the following clinical impacts. CE-TUS allows the specific identification of patients who are strongly suspected of having malignant pancreatic masses upon screening with TUS. CE-EUS facilitates the differential diagnosis of small pancreatic masses that are detected only on P-EUS, and it may lead to the early diagnosis of malignant pancreatic masses. CE-TUS/CE-EUS also has the following advantages over other modalities: CE-TUS/CE-EUS is advantageous when patients have contraindications to MRI and CT contrast agents such as renal failure or contrast allergy; the rate of adverse events with ultrasonography contrast agents is lower than that for iodinated contrast agents; ultrasonography itself is safer than CT because of the absence of ionizing radiation; and CE-TUS/CE-EUS allows for real-time dynamic and repeat examinations. Moreover, there is another role of contrast in the diagnostic algorithm of pancreatic lesions, for example, to direct the tissue sampling [42,62,63,64]. In a meta-analysis on EUS-FNA with CE-EUS (CE-EUS-FNA), the pooled diagnostic sensitivity was 84.6% (95% CI 80.7–88.6%) with CE-EUS-FNA and 75.3% (67–83.5%) with EUS-FNA, with evidence of a significant superiority of the former (OR 1.74, 95% CI 1.26–2.40; *p* < 0.001) [64]. Therefore, CE-EUS may contribute to an improved diagnostic performance of EUS-FNA in pancreatic lesions.

## 4. Conclusions

If this clinical trial demonstrates the efficacy and safety of CE-TUS/CE-EUS using perflubutane, it is likely to become an indispensable tool for the diagnosis of pancreatic malignant masses in clinical practice.

## Figures and Tables

**Figure 1 diagnostics-14-00130-f001:**
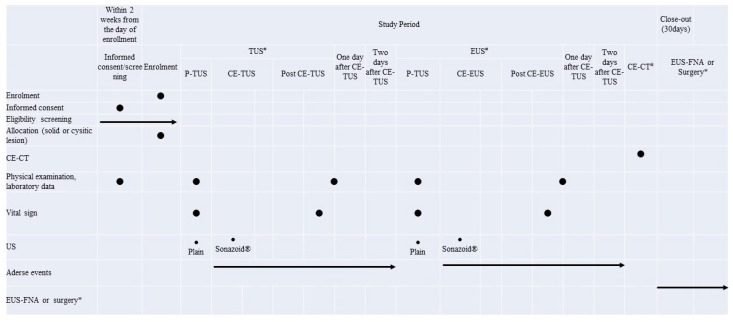
P-TUS, plain transabdominal ultrasonography; CE-TUS, contrast-enhanced transabdominal ultrasonography; P-EUS, plain endoscopic ultrasonography; CE-EUS, contrast-enhanced endoscopic ultrasonography; CE-CT, contrast-enhanced computed tomography; EUS-FNA, endoscopic ultrasonography-guided fine needle aspiration. * The final diagnosis is based on pathological findings obtained by EUS-FNA and/or surgery when pathological diagnosis is malignant. The final diagnosis is based on comprehensive diagnosis with imaging findings excluding CE-TUS/CE-EUS imaging, tumor markers, and pathological diagnosis when pathological diagnosis is not malignant. ^#^ The order of the three modalities does not matter, but no other test should be performed for 3 days after CE-CT and 2 days after CE-TUS or CE-EUS.

**Figure 2 diagnostics-14-00130-f002:**
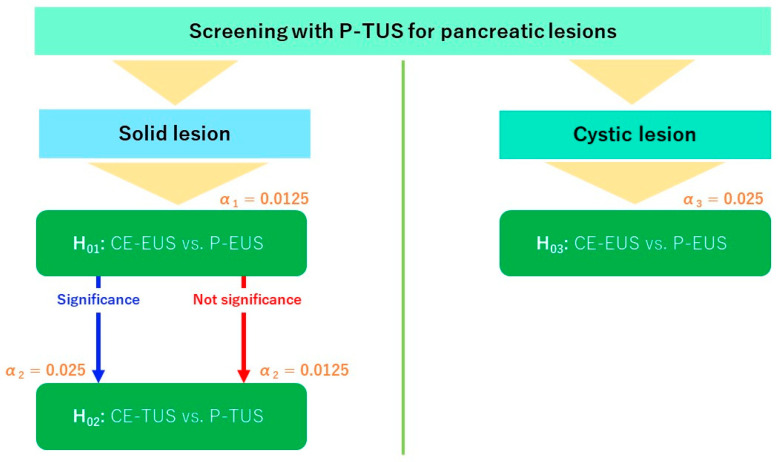
Statistics flowchart. P-TUS, plain transabdominal ultrasonography; CE-EUS, contrast-enhanced endoscopic ultrasonography; P-EUS, plain endoscopic ultrasonography; CE-TUS, contrast-enhanced transabdominal ultrasonography.

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
