# Peer review of "Protocol for a Multi-Center Confirmatory Trial to Evaluate the Differential Diagnostic Performance of Contrast-Enhanced Ultrasonography Using Perflubutane in Patients with a Pancreatic Mass: A Multicenter Prospective Study"

_diagnostics, 2024, doi:10.3390/diagnostics14020130_

Round 1

Reviewer 1 Report

Comments and Suggestions for Authors

Very interesting protocol for a RCT. My only minor comment is to improve the discussion mentioning other potential roles of contrast in the diagnostic algorithm of pancreatic lesions, for example to direct the tissue sampling (in this regard cite the recent MA: PMID: 33481633)

The authors should also comment on the contrast-enhancement characteristics of pancreatic masses (heterogeneity, necrotic areas, and so on....)

Author Response

Thank you for important comments. We have added the role of contrast for diagnosis with tissue sampling in the Discussion section.

Reviewer 2 Report

Comments and Suggestions for Authors

The study deals with a very important issue. However I feel confusions in the text. Among the abbreviation CE-US and CE-TUS are the same I think, please choose only one! Similarly EUS and P-EUS are the same. US and P-US, TUS are also the same terms, why the authors use several synonyms, This is also disturbing. In the Introduction CH-EUS is used instead of CE-EUS. I recommend to use one term for one investigation, like TUS, CE-TUS, EUS, CE-EUS. 

Author Response

Thank you very much for this pertinent comment. We apologize for the confusing description. We have unified the wording of US, CE-US, TUS, CE-TUS, EUS and CE-EUS as you have indicated.

Round 2

Reviewer 1 Report

Comments and Suggestions for Authors

The revised version of the manuscript is OK. Thank you!

Author Response

Thank you for your careful peer review.